# A randomized phase 3 trial of Gemcitabine or Nab-paclitaxel combined with cisPlatin as first-line treatment in patients with metastatic triple-negative breast cancer

Biyun Wang [1,13✉], Tao Sun[2,13], Yannan Zhao[1,13], Shusen Wang[3], Jian Zhang[1], Zhonghua Wang[4], Yue-E Teng[5], Li Cai[6], Min Yan[7], Xiaojia Wang[8], Zefei Jiang [9], Yueyin Pan[10], Jianfeng Luo[11], Zhimin Shao[4], Jiong Wu [4], Xiaomao Guo[12] & Xichun Hu [1✉]

Platinum is recommended in combination with gemcitabine in the treatment of metastatic triple-negative breast cancer (mTNBC). We conduct a randomized phase 3, controlled, open-label trial to compare nab-paclitaxel/cisplatin (AP) with gemcitabine/cisplatin (GP) in mTNBC patients (ClinicalTrials.gov NCT02546934). 254 patients with untreated mTNBC randomly receive AP (nab-paclitaxel 125 mg/m² on day 1, 8 and cisplatin 75 mg/m² on day 1) or GP (gemcitabine 1250 mg/m² on day 1, 8 and cisplatin 75 mg/m² on day 1) intravenously every 3 weeks until progression disease, intolerable toxicity or withdrawal of consent. The primary endpoint is progression-free survival (PFS); secondary endpoints are objective response rate (ORR), safety and overall survival (OS). The trial has met pre-specified endpoints. The median PFS is 9.8 months with AP as compared to 7.4 months with GP (stratified HR, 0.67; 95% CI, 0.50–0.88; $P = 0.004$). AP significantly increases ORR (81.1% *vs.* 56.3%, $P < 0.001$) and prolongs OS (stratified HR, 0.62; 95% CI, 0.44–0.90; $P = 0.010$) to GP. Of grade 3 or 4 adverse events, a significantly higher incidence of neuropathy in AP and thrombocytopenia in GP is noted. These findings warrant further assessment of adding novel agents to the nab-paclitaxel/platinum backbone due to its high potency for patients with mTNBC.

[1] Department of Breast Cancer and Urological Medical Oncology, Fudan University Shanghai Cancer Center, Department of Oncology, Shanghai Medical College, Fudan University, 270 Dong'an Road, Shanghai 200032, P.R. China. [2] Department of Medical Oncology, Cancer Hospital of China Medical University, Liaoning Cancer Hospital and Institute, No. 44 Xiaoheyan Road, Dadong, Shenyang, Liaoning 110042, P.R. China. [3] Department of Medical Oncology, Sun Yat-Sen University Cancer Center, The State Key Laboratory of Oncology in South China, Collaborative Innovation Center for Cancer Medicine, Guangzhou, Guangdong 510060, P.R. China. [4] Department of Breast Surgery, Fudan University Shanghai Cancer Center, Department of Oncology, Shanghai Medical College, Fudan University, Shanghai 200032, P.R. China. [5] Department of Medical Oncology, The First Hospital of China Medical University, Shenyang 110001, P.R. China. [6] Department of Medical Oncology, Harbin Medical University Cancer Hospital, Harbin 150081, P.R. China. [7] Department of Breast Disease, Henan Breast Cancer Center, The affiliated Cancer Hospital of Zhengzhou University & Henan Cancer Hospital, Zhengzhou 450000, P.R. China. [8] Department of Medical Oncology, Zhejiang Cancer Hospital, Hangzhou 310022, P.R. China. [9] Department of Medical Oncology, The Fifth Medical Center of Chinese PLA General Hospital, Beijing 100853, P.R. China. [10] Department of Medical Oncology, The First Hospital, Anhui Medical University, Hefei 230001, P.R. China. [11] Department of Biostatistics, School of Public Health, Fudan University, Shanghai 200032, P.R. China. [12] Department of Radiation Oncology, Fudan University Shanghai Cancer Center, Department of Oncology, Shanghai Medical College, Fudan University, Shanghai 200032, P.R. China. [13] These authors contributed equally: Biyun Wang, Tao Sun, Yannan Zhao. ✉email: wangbiyun0107@hotmail.com; xchu2009@hotmail.com

Triple-negative breast cancer (TNBC), lacks the expression of estrogen receptor (ER) and progesterone-receptor (PgR) and amplification of human epidermal growth factor receptor 2 (HER2) gene, accounts for up to 20% of all breast cancers, and is associated with aggressive behavior and poor prognosis[1]. Patients with TNBC are more likely to experience earlier relapse and shorter survival time as compared to those with other subtypes[2]. While endocrine therapy and HER2-targeted agents have significantly increased the survival benefit of luminal and HER2-positive metastatic breast cancer, the prognosis of TNBC remains poor. Although a variety of novel therapeutic strategies emerged in recent years, the median survival time for metastatic TNBC is still about 12–25.7 months[3–7].

TNBC shares molecular characteristics with basal-like breast cancer[8,9]. About 10–15% TNBCs carried BRCA mutation[10,11], while 47.7–71.0% harbored a deficiency in homologous recombination[6,12]. These phenomena provide a foundation for DNA cross-linking agents, such as platinum in the treatment of TNBC. The platinum treatment has aroused interest in early and advanced TNBCs, and evidence suggests that platinum-containing regimens are clinically beneficial to patients with TNBC. In the preoperative setting, the addition of platinum to neoadjuvant chemotherapy increased pathological complete response (pCR)[13,14], while prolonged progression-free survival (PFS) has also been observed among metastatic TNBC (mTNBC) patients treated with platinum-containing regimens[4,15,16]. Currently, platinum agents, carboplatin and cisplatin, have been recommended in combination with gemcitabine in the treatment of mTNBC[17–19].

Our previous study has proven the superiority of gemcitabine/cisplatin (GP) in first-line treatment of mTNBC in terms of PFS, compared with gemcitabine/paclitaxel (7.7 months for GP and 6.5 months for GT group, but no significant difference in OS)[15]. Thus, GP has been recommended in the guidelines of Chinese Breast Cancer Society and German Gynecological Oncology Working Group[18,19]. To further exploit the potential effect of cisplatin for TNBC, we chose nanoparticle albumin-bound (nab)-paclitaxel as the partner since it exerts the highest anti-tumor activity for metastatic breast cancer (MBC)[20]. Furthermore, the safety profile and activity of nab-paclitaxel/cisplatin (AP) were tested in patients with mTNBC in our previous prospective phase 2 study[21]. Based on the evidence of high response of nab-paclitaxel and synergistic effect of nab-paclitaxel and cisplatin, we hypothesized a superior efficacy of AP to GP as first-line treatment for patients with mTNBC, and the trial protocol was approved by Celgene Global Medical in 2015.

In this work, we report the results of GAP, a multicenter, randomized, open-label, first-line phase 3 trial to compare the efficacy of first-line AP to GP in mTNBC patients. We show an improved PFS with AP compared to GP as first-line treatment in patients with mTNBC.

## Results

**Patients**. A total of 254 patients were enrolled and randomly assigned to receive AP ($n = 127$) or GP ($n = 127$) between 30 March 2016 and 09 October 2019 (Fig. 1). One patient in the GP group did not receive the assigned treatment because of consent withdrawal. Thus, 253 patients, who received at least one dose of the study drugs (127 assigned to AP and 126 to GP), were included in the ITT and safety analysis. Patient demographics and baseline disease characteristics were well-balanced between the treatment groups (Table 1).

**Treatment exposure**. Patients received a median of six cycles of treatment in both AP and GP groups (range: 1–10 for the AP

group and 0–10 for the GP group). 241 (95.2%) of 253 patients underwent at least two cycles of treatment in this study. The relative dose intensity was 92 and 95% for nab-paclitaxel and cisplatin, respectively, and 92 and 94% for gemcitabine and cisplatin, respectively (Supplementary Table 1). At the cutoff point of the analysis (23 Feb 2021), the treatment has discontinued in both groups (Fig. 1). 26 (20.4 %) of 127 patients in the AP group and 47 (37.3%) of 126 patients in the GP group discontinued the treatment due to disease progression. Dose reduction occurred in 21 (16.5%) of 127 patients in the AP group and 47 (37.3%) of 126 patients in the GP group.

**Efficacy**. At the cutoff date of data analysis, a total of 202 patients, 101 in the AP group and 101 in the GP group, experienced disease progression or death. The median follow-up was 23.2 months for the AP group and 19.3 months for GP group in the whole population. PFS was significantly longer in the AP group than in the GP group (median, 9.8 months [95% CI 8.70–10.90] vs. 7.4 months [95% CI 5.93–8.94]; HR, 0.69; 95% CI 0.53–0.92; $P = 0.010$; stratified HR, 0.67; 95% CI 0.50–0.88; $P = 0.004$) (Fig. 2A).

For subsequent medical treatments after disease progression (Supplementary Table 5), 64 (63.4%) of 101 patients in the AP group and 54 (53.4%) of 101 patients in the GP group received subsequent treatments. Among them, 24 (37.5%) and 29 (53.7%) patients received vinorelbine-containing combination as second-line therapy in the AP and GP group, respectively. Moreover, 11 (9.5%) patients received bevacizumab, and 10 (18.5%) patients received a taxane-based treatment in the GP group.

Other key subgroups and exploratory analyses of PFS by stratification factors are shown in Fig. 3. The median PFS was significantly longer with the AP group than with the GP group in the majority of subgroups. In the ITT population, ORR, as assessed by the investigator, was 81.1% (103/127) in the AP group as compared to 56.3% (71/126) in the GP group ($P < 0.001$) (Table 2), while ORR by central assessment was 79.5% (101/127) and 58.7%, respectively ($P < 0.001$) (Supplementary Table 2). The median duration of response was 8.4 months (95% CI, 7.1–9.7) in patients treated with AP and 6.4 months (95% CI, 4.4–8.4) with GP (HR, 0.68; 95% CI 0.48– 0.95). At the time point of the data cutoff, 55 (44.3%) of 127 patients in the AP group and 67 (53.2%) of 126 in the GP group had died. OS was significantly different between two groups with the stratified HR of 0.62 (95% CI 0.44–0.90; $P = 0.01$; HR, 0.67; 95% CI 0.47–0.96; $P = 0.028$) and median OS was 26.3 months for the AP group and 22.9 months for the GP group (Fig. 2B).

**Safety**. All 253 patients in the safety population developed at least one AE (Table 3). All-grade AE that was at least 5% greater in the AP arm were neutropenia, neuropathy, anorexia, fatigue, nausea, and vomiting, and hypomagnesemia, however for grade ≥3 only neuropathy (19% vs 0%) and nausea and vomiting (6% vs 1%) appeared to markedly differ. Conversely, the GP arm had at least 5% more grade ≥3 thrombocytopenia (29.4% vs. 3.9%). Neutropenia, neuropathy, and anemia were the most commonly reported grade 3 or 4 AEs in the AP group, with a frequency of 40.9%, 18.9%, and 15.0%, respectively. The most commonly reported grade 3 or 4 AEs with GP were neutropenia (40.5%), thrombocytopenia (29.4%), and anemia (11.9%). SAEs were reported in 6 (4.7%) of 127 patients in the AP group and for 4 (3.2%) of 126 patients in the GP group. No treatment-related deaths were reported, while 2 patients died during the treatment of AP; one was due to the concurrent uncontrolled type 2 diabetes mellitus with sudden ketoacidosis, and the other was due to meningeal metastasis.

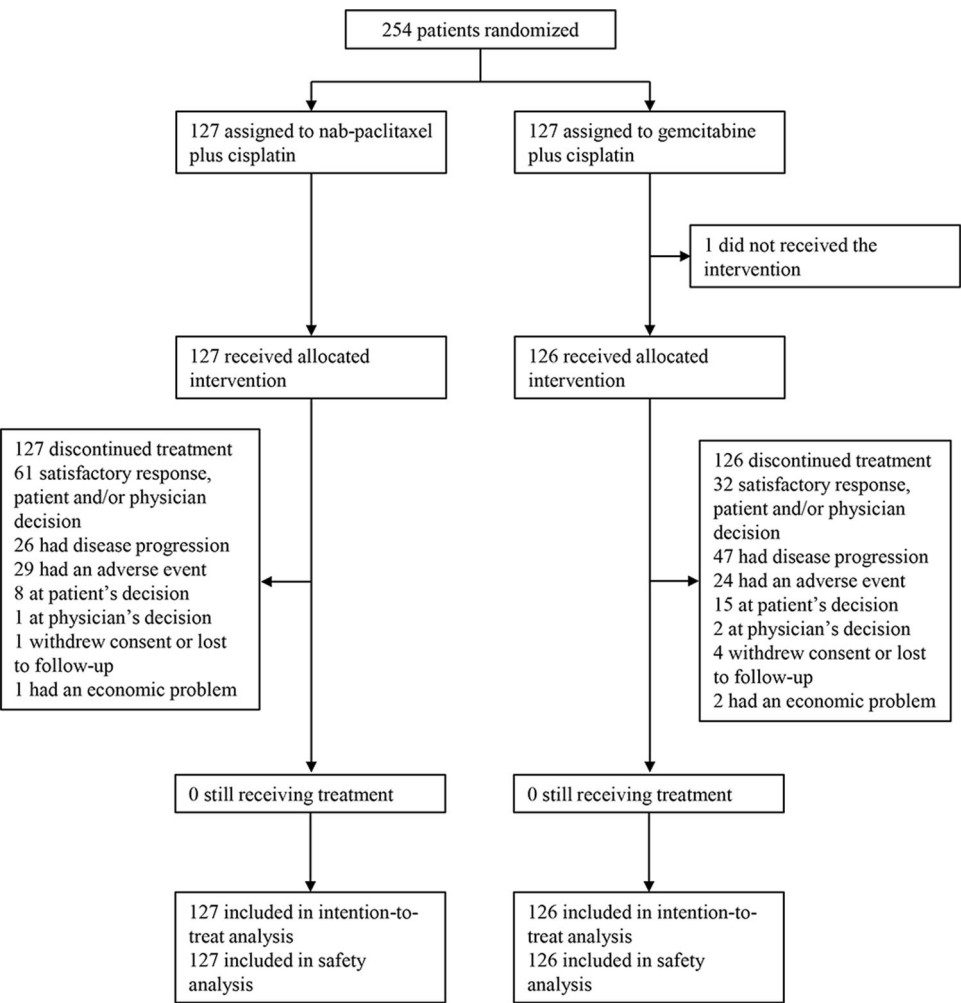

**Fig. 1 Consort diagram of the GAP study.** Patients may have discontinued the assigned treatment for multiple reasons. This figure shows the numbers of patients included in the intention-to-treatment and safety analyses.

## Discussion

As the first-line treatment, AP significantly improved PFS, as compared to GP in the ITT population with a reduction in the risk of progression or death by 34%. Importantly, we observed a 4.3 months increase in survival in AP group. AP also led to significantly higher ORR than that for gemcitabine and cisplatin. As expected, both combinations of the drugs were consistent with the known toxic effects of each agent.

To the best of our knowledge, AP in this trial reached the numerically longest PFS and highest response rate among the phase 3 studies in the first-line treatment of mTNBC, with the median PFS of 9.8 months and the ORR of 81.1%. Immune checkpoint inhibitors, such as atezolizumab, a PD-L1 antibody[3,22], and pembrolizumab, a PD-1 antibody[23], achieved clinically significant PFS and/or OS benefits (Supplementary Table 3). However, both were established on the basis of the relatively weaker control regimen with a median PFS of only 5.0–5.6 months[3,23], whereas GP doublet in this trial had a median PFS of 7.4 months. In addition, a quick response was realized with the progressive disease of only 0% (0/127), 2.4% (3/127), and 9.4% (12/127) at the end of cycles 2, 4, and 6, respectively (Supplementary Table 4). TNBC is an aggressive disease and usually requires a highly potent regimen to reduce the tumor burden and relieve the symptoms swiftly, which might in part be the reason for the success of the AP doublet. Lastly, the AP doublet was covered in the healthcare systems of China and may provide an affordable option for patients who are in poor economic conditions.

This study further confirmed the valuethe of first-line AP doublet. The efficacy of GP was similar in this trial to our previous phase 3 trial (7.4 and 7.73 months, respectively). This not only reaffirmed the anti-tumor activity of the GP regimen for mTNBC but based on the results from the two studies, a favorable efficacy of AP could be ascribed to paclitaxel plus gemcitabine by indirect comparison, further supporting the efficacy of AP regimen for patients with mTNBC. Nab-paclitaxel combined with carboplatin in the tnAcity trial, a randomized phase 2 trial, also confirmed the application of nab-paclitaxel and platinum for mTNBC[5]. Typically, for the biologically and molecularly similar cancer types with TNBC, such as serous ovarian cancer and lung squamous carcinoma[24], the doublet of taxane and platinum is the first-line standard of care. In TNBC, the addition of bevacizumab to nab-paclitaxel/carboplatin in a phase 2 trial reached a PFS of 9.2 months and an ORR of 85%[25], whereas adding pembrolizumab to paclitaxel/carboplatin followed by anthracycline plus cyclophosphamide in a phase 3 trial increased the pCR rate by 13.6%[26]. Therefore, it is reasonable to hypothesize that the first-line nab-paclitaxel/platinum doublet, rather than single-agent chemotherapy, may be the appropriate chemotherapy backbone to explore the addition of novel agents.

AP doublet translated its improvements in efficacy into a significant survival benefit. The reduction in risk of death may result

**Table 1 Patient characteristics.**

| Characteristics | Nab-paclitaxel plus cisplatin (n = 127) | Gemcitabine plus cisplatin (n = 126) | P value |
|---|---|---|---|
| Age (range) | 50 (22–69) | 52 (30–75) | 0.26 |
| <40 | 26 (20.5) | 19 (15.0) | |
| ≥40 | 101 (79.5) | 107 (85.0) | |
| ECOG performance status | | | |
| 0 | 18 (14.2) | 22 (17.5) | 0.47 |
| 1 | 109 (85.8) | 104 (82.5) | |
| Menopausal status | | | |
| Premenopausal | 61 (48.0) | 52 (40.9) | 0.28 |
| Postmenopausal | 66 (52.0) | 74 (58.7) | |
| Disease-free interval | | | |
| novo stage IV | 25 (19.7) | 23 (18.2) | 0.37 |
| <1year | 11 (8.7) | 18 (14.3) | |
| ≥1year | 91 (71.7) | 85 (67.5) | |
| Number of metastatic organ sites | | | |
| 1 | 44 (34.6) | 50 (39.7) | 0.65 |
| 2 | 41 (32.3) | 40 (31.7) | |
| ≥3 | 42 (33.1) | 36 (28.6) | |
| Metastatic sites | | | |
| Visceral disease | 86 (67.7) | 77 (61.1) | 0.27 |
| Lung | 57 (44.9) | 56 (44.4) | 0.94 |
| Liver | 34 (29.1) | 31 (24.6) | 0.69 |
| Bone | 37 (29.1) | 34 (27.0) | 0.70 |
| Lymph nodes | 87 (68.5) | 91 (72.2) | 0.52 |
| Neoadjuvant or adjuvant chemotherapy | | | |
| Anthracycline | 86 (67.7) | 87 (69.0) | 0.82 |
| Taxane | 86 (67.7) | 84 (66.7) | 0.85 |
| Anthracycline and taxane | 75 (59.0) | 75 (59.5) | 0.93 |
| Capecitabine | 2 (1.6) | 5 (4.0) | 0.44 |

*P value is given for chi-square test (two-sided) with no adjustment for multiple comparisons.*
*ECOG Eastern Cooperative Oncology Group.*

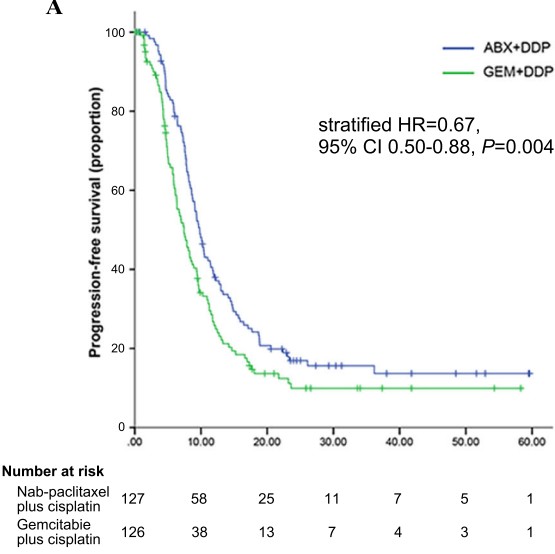

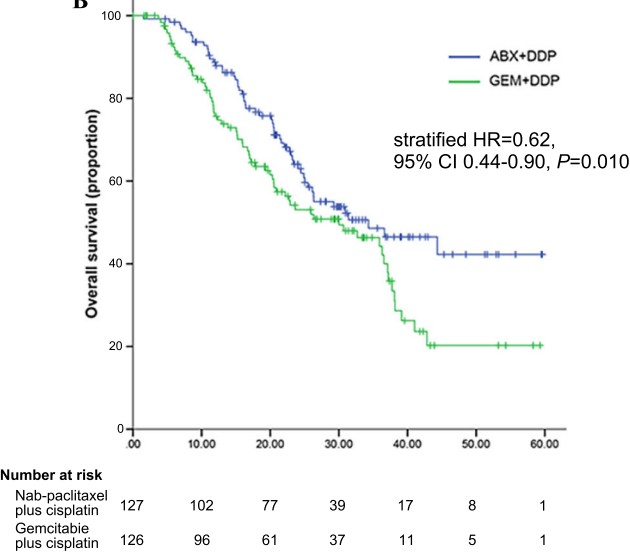

**Fig. 2 Kaplan–Meier estimates of survival curves for AP and GP groups.**
Kaplan–Meier plot of PFS (**A**) and OS (**B**). Data for the ITT population. Treatment effects were compared using the log-rank test; HRs and corresponding 95% CIs were estimated using Cox-proportional-hazard regression model. P values are one-sided with no adjustment for multiplicity. HR = hazard ratio.

from the higher anti-tumor activity of nab-paclitaxel over gemcitabine when combined with cisplatin. We first identified the survival benefit of chemotherapy doublet to another doublet in mTNBC population. Nevertheless, we don't know if AP may achieve the improved OS to nab-paclitaxel crossed over to cisplatin at the same doses and schedule.

Whether combination treatment or sequential treatment is optimal for mTNBC patients has been causing of discussion. Previous studies demonstrate that combination therapy brings no significant survival benefit and increased toxicity compared with sequential single-agent therapy, despite improving response rates and PFS[27]. However, the population of these studies included all subtypes, not limited to mTNBC. It is uncertain if mTNBC may benefit from a different approach due to its aggressive behavior with a higher tendency for heavy tumor burden and visceral metastasis. Furthermore, for first-line treatment of mTNBC, combination therapy is regularly considered both in clinical practice and clinical trials[5,23,28]. For example, in KEYNOTE-355 study[23], 55% of investigators chose gemcitabine plus carboplatin as the treatment of physician choice. For these reasons, we explored first-line combination chemotherapy for patients with mTNBC in our serial studies (NCT02341911)[15]. In this study, we provided more evidence for combination therapy and identified cisplatin as an alternative partner for combination therapy.

As expected, neuropathy rates were higher in the AP group than in the GP group (63% any-grade and 19% grade 3 for AP group), which could be attributed to nab-paclitaxel combined with concomitant neurotoxic cisplatin and the long exposure to both drugs. In our previous phase 2 trial, nab-paclitaxel (125 mg/m$^2$ on d1, d8

and d15) plus cisplatin (75 mg/m$^2$ on d1) every 28 days led to a peripheral neuropathy of 72.6% for any-grade and 26.0% for grade 3. Decreased relative dose intensity of nab-paclitaxel in this study seemed to reduce the incidence of peripheral neuropathy. Whereas, the tnAcity trial reported that grade 3 peripheral neuropathy of nab-paclitaxel plus carboplatin was 5% in 64 patients (the same dose of nab-paclitaxel with this study), indicating that cisplatin-induced peripheral neurotoxicity is more frequent and severe in conventional doses. Neuropathy should be considered when making decisions on the first-line treatment regimen. In general, most toxic effects of the AP were minor, rarely limited therapy. More patients experienced dose reduction in GP group than AP group (37.3% vs.16.5%) and patients who had treatment discontinuation were similar in the two groups (19.0% vs. 22.8%).

The present study also has some limitations. The sample size was evaluated based on the primary endpoint of PFS with 80%

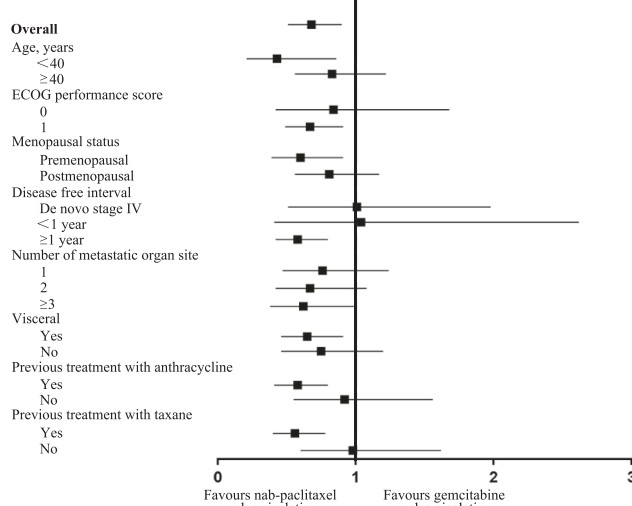

**Fig. 3 Subgroup analysis of PFS.** Data are median for ITT population in clinically relevant subgroups. Subgroup included patients with <40 years (*n* = 45) and ≥40 years (*n* = 208); ECOG = 0 (*n* = 40) and ECOG = 1 (*n* = 213); premenopausal status (*n* = 113) and postmenopausal status (*n* = 140); de novo stage IV (*n* = 48), disease-free survival <1 year (*n* = 29) and ≥1 year (*n* = 176); No. of metastatic organ site = 1 (*n* = 94), = 2 (*n* = 81) and ≥3 (*n* = 78); visceral metastasis (*n* = 163) and no visceral metastasis (*n* = 90); previous treatment with anthracycline (*n* = 173) and with no anthracycline (*n* = 80); previous treatment with taxane (*n* = 170) and with no taxane (*n* = 83). HRs and corresponding 95% CIs were estimated using Cox-proportional-hazard regression model. Data are presented as HR and 95% CI. *HR* hazard ratio, *ECOG* Eastern Cooperative Oncology Group.

**Table 2 Locally assessed response to study treatment in ITT population.**

| Response | ITT population | | |
|---|---|---|---|
| | ABX+DDP (*n* = 127) | GEM+DDP (*n* = 126) | *P* value |
| Complete response | 13 (10.2%) | 7 (5.5%) | |
| Partial response | 90 (70.9%) | 64 (50.8%) | |
| Stable disease | 20 (15.7%) | 40 (31.7%) | |
| Progressive disease | 0 (0%) | 7 (5.5%) | |
| Missing data or not assessable[a] | 4 (3.1%) | 8 (6.3%) | |
| Overall response | 103 (81.1%) | 71 (56.3%) | <0.001 |

[a]Tumor assessment data were missing or not assessable for response because of consent withdrawal before the first assessment in the ITT population.
*P* value is given for chi-square test (two-sided) with no adjustment for multiple comparisons.

power, which might be less powered for a phase 3 trial, and the sample size might not provide a full toxicity profile of AP regimen. A larger population is warranted to further identify the PFS and OS benefits of AP in mTNBC patients. Secondly, a retrospective translational study is being carried out, but we didn't obtain all samples from patients because the translational part had not been prospectively designed.

In conclusion, the AP doublet in patients with mTNBC, compared to GP, achieved greater efficacy with manageable toxicity, and hence, is an option as a first-line treatment. In the context of all the available first-line evidence, this doublet could also serve as the chemotherapy backbone to further assess novel agents due to the high potency for all-comers of mTNBC. A study

of AP combined with PD-1 antibody has been initiated in our center to further improve the efficacy and survival benefit of this regimen.

## Methods

**Study design and participants.** This open-label, randomized clinical trial was conducted at nine Chinese Breast Cancer Study Group (CBCSG) institutions or hospitals in China and registered as CBCSG018 study (Supplementary Table 6). The trial was registered with ClinicalTrials.gov, number NCT02546934. The first patient was enrolled on 30 Mar 2016, and the last patient was enrolled on 09 Oct 2019. The protocol of this study is available as Supplementary Note 2 in the Supplementary Information file. The eligible subjects were female patients aged 18–70 years, who had been diagnosed with mTNBC. ER, PgR, and HER2 status were determined locally by immunohistochemistry (IHC) of patients' primary or metastatic tumor sections. ER-negative and PgR-negative status was defined as ≤1% staining in the nuclei by IHC. HER2-negative status was defined by IHC staining 0 to 1+ or fluorescence in situ hybridization ratio <2.0 if IHC 2+ or IHC not performed. The metastatic disease was confirmed by clinical, imaging, histological, or cytological measures. Patients were not required to receive any previous chemotherapy or targeted therapy for mTNBC. Patients, who received adjuvant/neoadjuvant therapy, required an interval of at least 6 months between the last dose of the chemotherapy and the first documented time of disease release or distant metastasis. Eligibility criteria also included at least one measurable lesion according to Response Evaluation Criteria in Solid Tumors (RECIST) 1.1; a negative pregnancy test in women of childbearing potential; Eastern Cooperative Oncology Group (ECOG) performance status ≤1; life expectancy of >12 weeks, and adequate organ and bone marrow function.

Key exclusion criteria were as follows: Patient of childbearing potential but unwilling to receive contraception, radiation therapy of axial bone within 4 weeks before enrollment in the study, treatment with an experimental agent within the previous 4 weeks, symptomatic central nervous system (CNS) disease, other malignant diseases within the past 5 years (patients with basal cell skin carcinoma and cervical carcinoma in situ were allowed), and uncontrolled infection.

The study protocol, statistical analysis plan, and informed consent form were approved by the ethics committee of Fudan University Shanghai Cancer Center. All patients provided written informed consent, and the study was approved by the institutional review board at each participating center. The study was conducted in accordance with the Declaration of Helsinki.

**Random assignment and procedures.** Patients were randomly assigned in a 1:1 ratio to receive AP or GP. Randomization was carried out centrally with the use of block randomization of size eight and an interactive web-response system. The stratification factors included visceral metastasis (yes or no) and the number of metastatic sites (1, 2, or ≥3). The investigators or research coordinators (Supplementary Note 1) checked the inclusion and exclusion criteria and sent the random assignment forms by fax to the Clinical Research Coordination Office in Fudan University Shanghai Cancer Centre (Shanghai, China). The allocated treatment was sent back to the investigator by fax. The investigators, site personnel, and patients were not blinded to the treatment assignment.

Patients received either nab-paclitaxel (nab-paclitaxel 125 mg/m² on days 1 and 8; cisplatin 75 mg/m² on day 1) or gemcitabine plus cisplatin (gemcitabine 1250 mg/m² on days 1 and 8; cisplatin 75 mg/m² on day 1) intravenously every 3 weeks until disease progression according to RECIST 1.1, intolerable toxicity, treatment delay for >2 weeks, or patient's withdrawal of consent. Tumor assessment included computed tomography (CT) and/or magnetic resonance imaging (MRI) scan of the chest, abdomen, and brain at baseline and every two cycles until disease progression. The response was assessed according to RECIST 1.1 by local investigators and retrospective independent central radiologists blinded to the treatment. After treatment discontinuation, the survival status was obtained by telephone or at clinic visit every 3 months and would continue until death or loss to follow-up. The adverse events (AEs) were evaluated from the date of randomization, at the end of each chemotherapy cycle, and 28 days after the last dose of the study drugs.

**Outcomes.** The primary endpoint was locally assessed PFS, defined as the time from randomization to the first recorded occurrence of objective disease progression according to RECIST 1.1 or death from any cause. The secondary endpoints included objective response rate (ORR), overall survival (OS), and safety profile. AEs were graded using National Cancer Institute Common Terminology Criteria for Adverse Events version 4.0. The correlation between the AEs/serious AEs (SAEs) and the treatment was determined by the investigators. Any AE leading to the dose reduction, dose delay, dose miss, or discontinuation was also recorded.

**Statistical analysis.** Based on our previous phase 2 trial on AP[21], we observed a median PFS of 10.3 months in the mTNBC subgroup. We expected that AP improved the median PFS from 7.73 months of GP combination, based on our previous phase 3 study, to 11.5 months in patients with untreated mTNBC. A total of 115 patients were required with 80% power using a log-rank (Lakatos) test at a two-sided 5% level of significance, with a 4-year accrual, 1-year follow-up. Considering a 10% dropout

**Table 3 AEs.**

| AEs | ABX+DDP (n = 127) | | | | GEM+DDP (n = 126) | | | |
|---|---|---|---|---|---|---|---|---|
| | All grade (n/%) | Grade 3 (n/%) | Grade 4 (n/%) | Grade 3/4 (n/%) | All grade (n/%) | Grade 3 (n/%) | Grade 4 (n/%) | Grade 3/4 (n/%) |
| Anemia | 114 (89.7) | 19 (15) | 0 (0) | 19 (15) | 107 (84.9) | 14 (11) | 1 (0.8) | 15 (11.9) |
| Neutropenia | 121 (95.3) | 35 (27.6) | 17 (13.4) | 52 (40.9) | 109 (86.5) | 36 (28.3) | 15 (11.8) | 51 (40.5) |
| Febrile neutropenia | 6 (4.7) | 6 (4.7) | 0 (0) | 6 (4.7) | 4 (3.2) | 4 (3.2) | 0 (0) | 4 (3.2) |
| Thrombocytopenia | 20 (15.7) | 4 (3.1) | 1 (0.8) | 5 (3.9) | 66 (52.4) | 21 (16.5) | 16 (12.6) | 37 (29.4) |
| Neuropathy | 80 (63.0) | 24 (18.9) | 0 (0) | 24 (18.9) | 22 (17.5) | 0 (0) | 0 (0) | 0 (0) |
| Diarrhea | 18 (14.2) | 1 (0.8) | 0 (0) | 1 (0.8) | 3 (2.4) | 0 (0) | 0 (0) | 0 (0) |
| Anorexia | 26 (20.5) | 1 (0.8) | 0 (0) | 1 (0.8) | 10 (7.9) | 0 (0) | 0 (0) | 0 (0) |
| Fatigue | 44 (34.6) | 0 (0) | 0 (0) | 0 (0) | 37 (29.4) | 1 (0.8) | 0 (0) | 1 (0.8) |
| Rash | 23 (18.1) | 0 (0) | 0 (0) | 0 (0) | 17 (13.5) | 0 (0) | 0 (0) | 0 (0) |
| Blurred vision | 6 (4.7) | 0 (0) | 0 (0) | 0 (0) | 4 (3.2) | 0 (0) | 0 (0) | 0 (0) |
| ALT/AST increased | 20 (15.7) | 1 (0.8) | 0 (0) | 1 (0.8) | 23 (18.2) | 0 (0) | 0 (0) | 0 (0) |
| Nausea or vomiting | 84 (66.1) | 8 (6.3) | 0 (0) | 8 (6.3) | 75 (59.5) | 1 (0.8) | 0 (0) | 1 (0.8) |
| Edema | 5 (3.9) | 0 (0) | 0 (0) | 0 (0) | 0 (0) | 0 (0) | 0 (0) | 0 (0) |
| Myalgia | 18 (14.2) | 0 (0) | 0 (0) | 0 (0) | 13 (10.3) | 0 (0) | 0 (0) | 0 (0) |
| Creatinine increased | 20 (15.7) | 0 (0) | 0 (0) | 0 (0) | 17 (13.5) | 0 (0) | 0 (0) | 0 (0) |
| Hypomagnesaemia | 35 (27.5) | 2 (1.6) | 0 (0) | 2 (1.6) | 27 (21.4) | 0 (0) | 0 (0) | 0 (0) |
| Hypokalemia | 21 (16.5) | 2 (1.6) | 1 (0.8) | 1 (0.8) | 17 (13.5) | 0 (0) | 0 (0) | 1 (0.8) |
| Hyponatremia | 24 (18.9) | 4 (3.1) | 0 (0) | 4 (3.1) | 22 (17.5) | 1 (0.8) | 0 (0) | 1 (0.8) |
| Hypocalcemia | 22 (17.3) | 0 (0) | 0 (0) | 0 (0) | 23 (18.2) | 1 (0.8) | 0 (0) | 1 (0.8) |

*ALT* alanine aminotransferase, *AST* aspartate aminotransferase. Data are *n* (%) from the safety population. There were no grade 5 drug-related AEs.

rate and 1:1 randomization, a total of 254 patients were required, with 127 patients per group. The primary analysis was carried out on an intention-to-treat (ITT) population, defined as all the patients who were treated with at least one dose of any study drug. Safety analysis was performed in the same population. All of the alpha of 0.05 was assigned to assess the primary endpoint of PFS.

We presented PFS and OS with the median time to the event using Kaplan–Meier plots. Hazard ratios and corresponding 95% confidence intervals (CIs) were assessed by the Cox-proportional-hazard regression model. Stratified HR and 95% CIs were calculated using the factors applied for randomization. Descriptive statistics were used to summarize the characteristics and AEs of the patients. The ORR was calculated by the treatment group with 95% CIs and compared using the chi-square test. Also, a posthoc subgroup analysis was done in the ITT population to compare the treatment effect of two groups in patients with different baseline characteristics including age (<40 or ≥40 years), ECOG (0 or 1), menopausal state (premenopausal or postmenopausal), disease-free survival (de novo stage IV, <1 or ≥1 year), previous treatment with anthracycline (yes or no), and previous treatment with taxane (yes or no). HRs and 95% CIs were calculated using a Cox-proportional-hazard model to assess whether the treatment effect was consistent across the factors. All statistical analyses were carried out using SPSS software (version 19.0, SPSS Inc., Chicago, IL, USA).

**Reporting summary.** Further information on research design is available in the Nature Research Reporting Summary linked to this article.

## Data availability

The study protocol and statistical analysis plan are available as Supplementary Note 2 in the Supplementary Information file. Data collected for the study, including individual participant data and a data dictionary defining each field in the set, will be made available to others; these data are available under restricted access in compliance with patient consent and ethical principles for data sharing. Access can be obtained by contacting corresponding author Xichun Hu (xchu2009@hotmail.com) with a scientific proposal including objectives. The data will be shared after approval by Xichun Hu and by the investigators of the GAP trial within two weeks. All data shared will be de-identified. The remaining data are available within the Article and Supplementary Information.

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

## Acknowledgements

Nab-paclitaxel was provided by Celgene Corporation. We thank all the patients, the GAP Study investigators, and their institutions for the time and effort put into this study. This study was funded by the National Natural Science Foundation of China, grant number 81874114 (for B.W.).

## Author contributions

Conception and design: X.H., B.W. Provision of study material or patients: X.H., B.W., T.S., S.W., J.Z., Z.W., Y.T., L.C., M.Y., X.W., Z.J., and Y.P. Collection and assembly of data: X.H., B.W., T.S., Y.Z., S.W., J.Z., Z.W., Y.T, L.C., M.Y., X.W., Z.J., and Y.P. Data analysis and interpretation: X.H., B.W., Y.Z., J.L. Manuscript writing: X.H., B.W., Y.Z. Final paper editing: X.G., Z.S., and J.W. Final approval of manuscript: All authors.

## Competing interests

The authors declare no competing interests.
