## [Peer Review File · Nature Communications]

A randomized phase 3 trial of Gemcitabine or Abraxane combined with cisPlatin as first-line treatment in patients with metastatic triple-negative breast cancerEditorial Note: This manuscript has been previously reviewed at another journal that is not operating a transparent peer review scheme. This document only contains reviewer comments and rebuttal letters for versions considered at Nature Communications.

REVIEWERS' COMMENTS

Reviewer #2 (Remarks to the Author):

One minor comment in Statistical Analysis of Method section.

“Also, a post-hoc subgroup analysis was done in the ITT population to compare the efficacy in patients with different baseline characteristics ..”.

The comparison seems to evaluate treatment assignment, not efficacy.

Reviewer #3 (Remarks to the Author):

The authors have adequately addressed my comments.

Point-by-point Responses

We are very grateful for the reviewer's insightful comments and suggestions.

Reviewer #2

Remarks to the Author:

One minor comment in Statistical Analysis of Method section. “Also, a post-hoc subgroup analysis was done in the ITT population to compare the efficacy in patients with different baseline characteristics ..”. The comparison seems to evaluate treatment assignment, not efficacy.

Thank you for your valuable advice. Actually, we compared PFS of two groups through a post-hoc subgroup analysis in patients with different baseline characteristics and HRs were used to calculate the treatment effect between two groups (Figure 3). To make it more clearly to understand, we revised it as “Also, a post-hoc subgroup analysis was done in the ITT population to compare the treatment effect of two groups in patients with different baseline characteristics.” **(line 365)** .